# Examining the Safety Impacts of High-Occupancy Vehicle Lanes: International Experience and an Evaluation of First Operation in Israel

Victoria Gitelman [1,*] and Etti Doveh [2]

1   Transportation Research Institute, Technion–Israel Institute of Technology, Technion City, Haifa 32000, Israel
2   Technion Statistical Laboratory, Technion–Israel Institute of Technology, Technion City, Haifa 32000, Israel; ierde01@technion.ac.il
*   Correspondence: trivica@technion.ac.il

**Abstract:** Current transport policies promote better use of existing roadways by using traffic management strategies such as high-occupancy vehicle (HOV) lanes. International experience showed positive mobility impacts of HOV lanes, while research evidence on their safety implications is limited. In Israel, the first HOV lanes were introduced in 2019. This study examined the impacts of HOV lanes on road safety based on a detailed review of international research and accident analyses, which evaluated the safety effects of HOV lanes in Israel. The literature survey applied a systematic screening of research studies from the past two decades and found that HOV lanes were frequently associated with an adverse effect on road safety. Yet, findings were limited to the North American experience, with mostly left-side HOV lanes in use, while in Israel, right-side HOV lanes were introduced. In Israeli evaluations, before-after comparisons of accident changes with comparison groups were applied, with regression models fitted to monthly time series of 17 accident types. Results showed that HOV lanes' operation led to increasing accident trends, particularly in interchange areas and in the daytime. In injury accidents on road sections, an average increase of 31–41% was found (yet non-significant), while at interchange areas, an increase was even higher and sometimes significant. Thus, adverse safety effects should be expected and accounted for in future planning of HOV lanes. Further research should explore the design features of HOV lanes to reduce their negative safety implications.

**Keywords:** safety; high-occupancy vehicle lanes; motorway; accidents; before-after evaluation

## 1. Introduction

Following continuous growth in population density and motorization levels worldwide [1], traffic congestion has been the bane of urban/suburban road networks for decades [2]. For example, in the USA, the annual congestion costs were estimated at $121 billion [3]. In the past, the problem was "solved" by building more roads. In light of increasing awareness of transport systems' financial, social, and environmental implications current prospective thinking focuses on better use of existing roadways, promoting the use of public transport and traffic management strategies such as high-occupancy vehicle (HOV) lanes [4–6].

HOV lanes were among the first managed lane concepts that were developed in the world; they were originally applied in the USA and later on in other countries [3,5–7]. HOV eligibility restricts lane use to vehicles with a minimum number of persons traveling in each vehicle (e.g., two or more); access can be restricted to specific access/egress points to manage demand and enable better traffic flow or not restricted. HOV lanes provide the benefit of a faster-flowing lane than the adjoining general-purpose, congested lanes, encourage car-pooling, and thus result in a higher per-lane person throughput than general-purpose lanes [3,6,7]. This way, HOV lanes allow more efficient use of the roadway, on the

one hand, as well as travel-time savings and better reliability for high-occupancy travel modes, on the other hand.

In the USA and Canada, HOV lanes are by far the most common concept of managed lanes; for example, according to the US handbook [8], HOV lanes are present on about two thousand North American freeway route-kilometers, while, according to another source, in California, the statewide HOV system has grown to over 2200 directional lane-kilometers [9]. Examples of implemented HOV lanes can also be found in Great Britain, the Netherlands, Spain, Norway, and Australia [5,7,10]. In practice, various forms of HOV lanes exist with regard to eligibility restrictions, e.g., vehicle types, number of vehicle occupants, time periods for operation and road toll, physical separation of HOV facilities, road design, etc. [3,4,7,8]. Examples from the US States, Canada and Spain showed the efficiency of HOV lanes relative to general-purpose lanes in terms of carrying a higher number of people in fewer vehicles, increasing vehicle occupancy, travel-time savings, higher use of public transport and a reduction in single-occupied cars' volume, on the route [4–8].

In Israel, the concept of HOV lanes has been promoted and introduced recently, aiming to enhance the efficiency of public transport routes, which are extensively planned today in the metropolitan areas of the country, within the so-called "Rapid to the City" program [11]. The reasons for adding HOVs to public transport routes lie in providing better use of road capacity and reducing public objections due to the "empty lane syndrome" [5,7]. To promote the effective use of HOV lanes under local conditions, guidelines were drafted with regard to implementation criteria and conditions for the use of HOV lanes, design rules for their settings, signing and marking [12]. The Israeli guidelines were developed based on a summary of the international experience [13], focusing on justification criteria for HOV lanes' implementation, traffic and road considerations, design settings and performance indicators applied in other countries [4,8,14,15].

The first HOV lanes were activated in Israel in October 2019 on an interurban motorway, Road #2. Those are right-side lanes in the road layout dedicated to "2+" HOVs, with a double yellow painted stripe separation from other lanes (Figure 1). The HOV lanes' introduction was accompanied by an evaluation study conducted in cooperation between the Ministry of Transport, the National Transport Infrastructure Company, and researchers [16]. The results indicated generally positive impacts of the measure on road capacity in terms of a higher number of buses using the HOV lanes, more fluent traffic conditions and sufficiently high travel speeds. However, safety concerns were raised regarding the HOV lanes' operation, particularly in merging and diverging areas near interchanges, where HOV lanes are crossed by many vehicles which need to leave or enter the motorway through the HOV lane. Such concerns relied on general safety knowledge that higher frequencies of vehicle interactions may increase the risk of accidents [17,18] and that merging and diverging areas near interchanges (still without HOV lanes' settings) are associated with higher accident risks than other road sites [19,20].

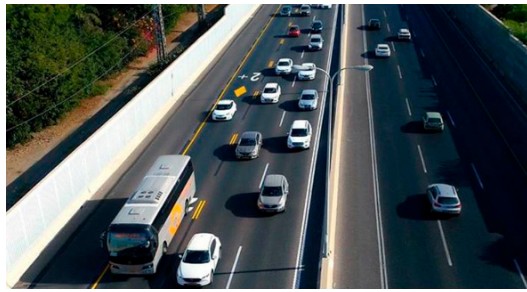

**Figure 1.** A view of HOV lanes in Israel, Road #2.

Concerning the safety impacts of HOV lanes, a literature survey that summarized the international experience and served as a basis for the local guidelines [13] indicated that previous examinations of accident changes related to HOV lanes were not frequent,

mostly performed in the USA and, sometimes, reported inconsistent results. For example, Cooner and Ranft [21] examined the impacts of HOV lanes on freeways in Texas and found an increase in crash rates after their implementation. In contrast, Pearce and Stanek [22] examined the safety performance of two freeways in California and did not find a consistent increase in accident rates following the HOV lanes' operation. Based on earlier studies in the USA, the Handbook of Road Safety Measures–Elvik et al. [23] reported summary estimates of an increase in accidents associated with HOV lanes, i.e., an average addition of 12% in injury and 15% in damage-only accidents (both changes are significant). Similarly, Cooner and Ranft [21] reviewed earlier US studies, from the eighties and nineties, regarding the safety of HOV lane projects and concluded that those have been relatively inconclusive due to data limitations in quality and quantity. Some studies did not find an adverse effect on the safety of the corridor with HOV lanes, while others raised safety concerns, mostly related to the speed differentials between the HOV and the general-purpose lanes [21].

Recognizing the positive impacts of HOV lanes on the effective use of road network capacity and a shift to public transport and shared trips [4–8,13–15], this study aimed to examine the impacts of HOV lanes on road safety. To attain its aim, the study included two components: a detailed review of the international literature on the safety impacts of HOV lanes while focusing on publications from the last two decades and accident analyses to evaluate the safety effects of HOV lanes that were introduced in Israel, in the first period of their operation. Both components intended to reduce existing research gaps since an updated summary on the issue was not available in the previous literature while regarding the safety impacts of right-side HOV lanes (as were applied in Israel), research findings were generally missing (see Section 2). Furthermore, the study produced new estimates of the safety impacts of HOV lanes in an additional country (Israel) where the HOV lanes' application policy is emerging, thus extending international knowledge on the topic. From a policy perspective, the study examinations were needed to support an understanding of the safety implications of the current developments in the transportation system. This can also be relevant to the "road safety impact assessment" of transport projects, as recommended by the European directive on road infrastructure safety management [24].

The remainder of the paper is structured as follows. Section 2 provides findings of the international research literature regarding the safety impacts of HOV lanes. Section 3 describes the methodology applied to analyse accident changes on roads with HOV lanes in Israel. Section 4 shows the evaluation results. Section 5 discusses the study findings in the context of international research and current knowledge needs. Section 6 suggests the main study conclusions for transport development practice.

## 2. Previous Research on Safety Impacts of HOV Lanes

### 2.1. Literature Search

The literature survey included a systematic screening of research studies following common guidelines [25,26] and learning from examples of their applications, see [27,28]. The screening methodology required a formulation of the problem and research questions; definition of a source search strategy, including processing multiple channels; preliminary evaluation and a further check of retrieved sources, including the selection criteria of suitable data; analysis and interpretation of the literature sources selected, and providing summary results as to the raised research questions.

In this study, the literature survey intended to review research studies which examined the safety impacts of HOV lanes during the past two decades. Thus, the literature search included studies published between 2001–2022 and applied keyword combinations such as: "safety impact *and* high-occupancy vehicle lane", "crash prediction *and* freeway *and* high occupancy vehicle lane"; "crash freeway HOV lane"; "accident freeway HOV lane". (The terms "accident" and "crash" are applied interchangeably in this study as both terms are common in the road safety literature.) The papers' search was based on the title, abstract and keywords and included several steps. A basic search was conducted using two databases commonly applied for screening road safety research: *TRID* and

*Scopus*. The TRID (Transport Research International Documentation) is a research database that combines records from the Transportation Research Board's Transportation Research Information Services focusing on transportation research. Scopus is Elsevier's research database, which covers, among others, the areas of social sciences and health science. Complementary searches were conducted on *Science Direct*, *Springer Link* and *Google Scholar* databases. Such an approach was applied, for example, by a European project, SafetyCube, which conducted standardized literature searches to update safety effects for a variety of risk factors and safety-related measures [29] and established a decision-support system with summary results [30].

Figure 2 illustrates the process of identification and selection of sources for the review in this study. The literature search was conducted in April 2022. The aforementioned databases (*TRID*, *Scopus* and others) were searched to identify published articles and scientific reports using the keywords defined; initially, 94 records were found. Three records were added from other sources, e.g., those collected during the preparation of the local design guidelines [13]. All retrieved abstracts were screened, and sources irrelevant to the study scope were excluded. Since multiple searches were conducted, the records were checked for duplicates. After screening abstracts and removing duplicates, 21 sources remained.

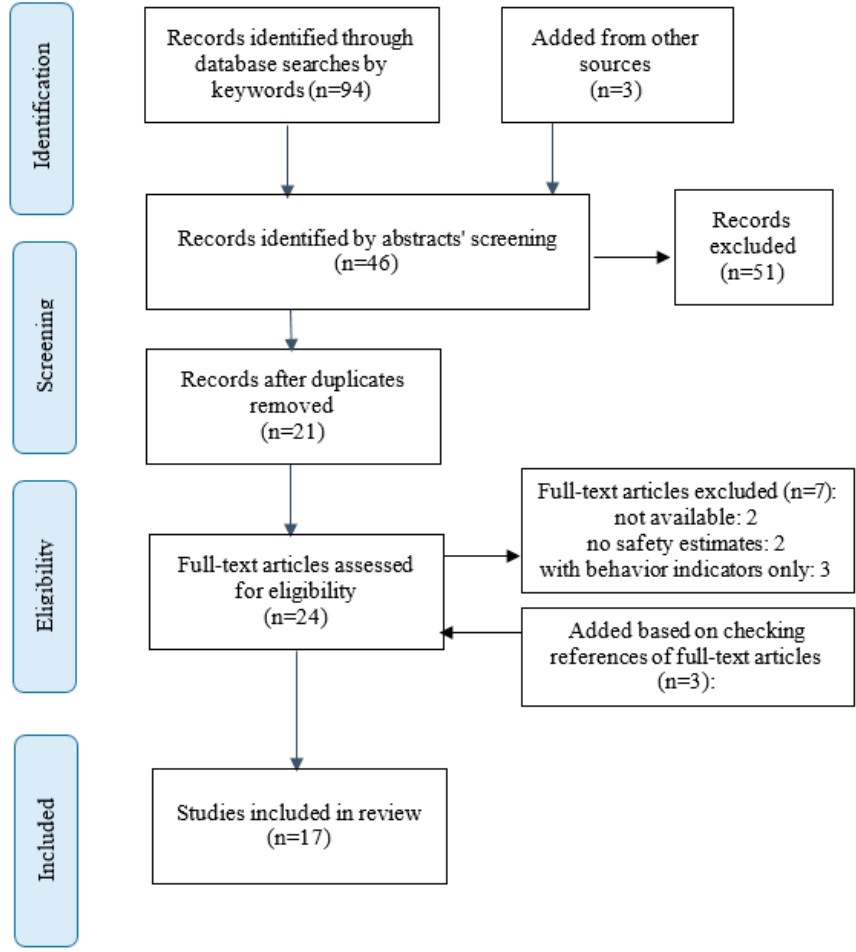

**Figure 2.** Flow diagram to identify the sources for the review.

Furthermore, full-text articles were retrieved and screened for relevance for the survey purposes. At this step, seven sources were excluded due to various reasons such as the unavailability of the full-text article (2), lack of safety estimates in the paper (2) or an evaluation of behavior indicators only (3), see Figure 2. On the other hand, three additional sources were found, having checked the references of the full-text articles.

After screening the identified sources and reviewing full texts, 17 sources were included in this review. All these studies examined accident changes or occurrences associated with HOV lanes' operation. As mentioned above, there were three additional studies which were checked in the screening process but excluded from the final review, as they examined HOV lanes' safety impacts in terms of vehicle behaviors and not accidents [31–33]. Two of them used simulation models and counted vehicle conflicts [31,32]. One study conducted observations on freeway sites to estimate the shares of vehicle lane change maneuvers to and from the HOV lanes under various traffic volumes and travel speeds [33].

### 2.2. Main Findings of the International Literature

The review of international literature included 17 studies with accident analyses of HOV lanes' operation [21,34–49]. To provide an effective summary of previous research, each study content was coded with regard to the features, such as the measure examined—HOV lane types; road characteristics considered—road classes and site types; data samples involved; study design and evaluation methods; safety indicators estimated and main results. Summary characteristics of the studies reviewed are presented in Table 1.

As can be seen in Table 1, all the reviewed studies were conducted in North American countries: 16 in the USA and one in Canada. All studies considered HOV lanes' operation on freeways, with mainly left-side HOV lanes (adjacent to the median), which are common in North American practice. Concerning the safety impacts of HOV lanes, the main findings can be summarized as follows (see studies in Table 1).

The safety effects of HOV lanes were estimated by means of analysis of descriptive accident statistics, cross-sectional or before-after evaluations, or fitting explanatory models for accident occurrences (typically, negative-binomial regression models). Many studies found an increase in crashes following the introduction of HOV lanes (in after-before comparisons) or associated with the presence of HOV lanes in the roadway layout (in explanatory models). For example, in several studies with after-before comparisons in Texas [21,34,35], an increase in crashes was observed following the introduction of HOV lanes. The increases in crashes were attributed to speed differentials between the HOV and general-purpose lanes, the reduced road cross-section [21,35] as well as to conflicts at intermediate access locations and lane changes by illegal users of the HOV lanes [34].

Similarly, studies in California and Canada [37,44] reported an increase in accidents. In California, a 10–11% increase in collisions was observed after the freeways were changed: the inside shoulder was converted to a concurrent flow HOV lane, and the other lanes were reduced in width [37]. In Canada, following the addition of limited-access HOV lanes to freeways, in the Greater Toronto and Hamilton Area, a 19% increase in damage-only crashes and a 15% increase in total crashes was found (both are significant), yet with no change in fatal and injury crashes [44]. Furthermore, a significant increase was reported in rear-end collisions, particularly within the weaving/merging zones of the HOV lanes [44].

However, there were several studies which did not find significant impacts of HOV lanes on safety. For example, in Texas, barrier-separated HOV lanes did not have an effect on injury crash rates [35]. In Utah, in the first two years of HOV lanes' operation (in the Salt Lake Valley), the accident statistics indicated no adverse effect on safety conditions, yet a before-after analysis was not applied due to substantial changes in the road layout [36]. In Virginia, Lee et al. [38] examined a setting which included left-side HOV lanes, with no separation, and the use of right shoulders in peak hours and found no significant impact of HOV lanes on crash frequency.

**Table 1.** Summary of previous research on HOV lanes' safety performance.

| Source | Country, Area | Measures' Description—HOV Lanes | Road Class, Sites | Crash Data Periods | Study Design | Other Road/Traffic Features Considered | Main Findings |
|---|---|---|---|---|---|---|---|
| Skowronek et al. (2002) [34] | Texas, USA | Left, buffer-separated and barrier-separated | Three freeway corridors of 6–8 mi each | FI crash data for 3–5 years in before and after periods | BA comparison of crash rates *; comparison with critical crash rates for similar corridors | --** | <ul><li>Crash rates increased after the implementation of buffer-separated HOV lanes, particularly in peak periods. Yet, HOV lanes' crash rates were comparable to similar freeway corridors.</li><li>The crash rate increase was partly attributed to conflicts at intermediate access locations and lane changes by illegal users of the HOV lane as they approached enforcement areas.</li><li>Higher crash rates on barrier-separated HOV lanes were related to construction projects and not to the HOV facilities.</li></ul> |
| Cothron et al. (2004) [35] | Texas, USA | Left contraflow, separated with a moveable barrier | Freeway corridor, 5.6 mi | Injury crash data for 6–9 years in each period | BA comparison of crash rates * | -- | <ul><li>No change in injury crash occurrence.</li></ul> |
| Cooner, Ranft (2006) [21]; Cothron et al. (2004) [35] | Texas, USA | On the left shoulder, with a painted buffer | Two freeway corridors of 6–7 mi each | Injury crash data for 4–5 years in each period, by severity; police crash reports (1150) | BA comparison of crash rates *; descriptive statistics of police reports | Buffer width, shoulder presence, lane width, speed differential between HOV and GP lanes | <ul><li>A 41–56% increase in crash rates after HOV lanes' implementation.</li><li>The increase in crashes was mainly on the HOV lanes and the first adjacent GP lanes.</li><li>The crash increase was attributed to speed differentials between the HOV and GP lanes and the reduced road cross-section.</li></ul> |
| Martin et al. (2004) [36] | Utah, USA | Left, with a painted separation | Freeway, 16 mi | 2.5-year crash records, in after period | Descriptive statistics * | -- | <ul><li>No definite trend was reported as to accident rates or severity.</li></ul> |

<div style="text-align: center;">**Table 1.** *Cont.*</div>

| Source | Country, Area | Measures' Description—HOV Lanes | Road Class, Sites | Crash Data Periods | Study Design | Other Road/Traffic Features Considered | Main Findings |
|---|---|---|---|---|---|---|---|
| Bauer et al. (2004) [37] | California, USA | The inside shoulder converted to an HOV lane, with painted separation; other lanes reduced in width | Urban freeways, 490 sites, 247.6 mi in total | 2-year before, 7-year after crashes | BA evaluation, with EB | -- | • An increase of 10% to 11% in collision frequency, presumably due to collision migration caused by the relocation of traffic operational bottlenecks. |
| Lee et al. (2007) [38] | Virginia, USA | Left, no separation, and use of right shoulders as GP lanes in peak hours | Urban freeway, 6.5 mi | 3-year crash data | NB regression models for the daily number of crashes | AADT, adverse weather, adverse light conditions | • No significant impact of HOV lanes' operation on crash frequency on the inner lanes. |
| Chung et al. (2007) [39] | California, USA | Left, buffer-separated and with continuous access | Eight freeway corridors, 78 mi in total | 10-year crash data | Descriptive statistics–crash patterns | -- | • Rare-end and side-swipe collisions on the HOV lanes were higher in limited-access corridors.<br>• On HOV lanes, rear-end and sideswipe crashes accounted for 75–90% of total crashes.<br>• HOV and left lanes had a greater concentration of collisions in peak hours. |
| Jang et al. (2009a) [40]; Jang et al. (2009b) [41] | California, USA | Left, with and without painted separation | Eight freeway corridors, 92 mi in total | 5-year crash data during peak hours | Descriptive statistics; cross-section comparison | Shoulder width, length of access, the proximity of access to neighboring ramps | • Limited access HOV lanes showed higher PDO and injury crash rates (per exposure) than those with continuous access.<br>• HOV facilities with continuous access had 16% fewer fatal and injury crashes than those with limited HOV access.<br>• HOV lanes' crash rates diminish with an increase in shoulder width, regardless of the type of access.<br>• Limited access HOV facilities with a short ingress/egress length and a close proximity to on- or off-ramp showed higher crash rates. |

**Table 1.** *Cont.*

| Source | Country, Area | Measures' Description—HOV Lanes | Road Class, Sites | Crash Data Periods | Study Design | Other Road/Traffic Features Considered | Main Findings |
|---|---|---|---|---|---|---|---|
| Cao et al. (2012) [42] | Minnesota, USA | Conversion of HOV to HOT lanes with adding access points; left reversible lanes, with painted separation | Urban freeway, 239 road segments | 4-year before, 2-year after crashes | BA evaluation, with fitting SPF | AADT, number of ramps | • A 5.3% reduction observed compared to expected crash numbers (by SPF).<br>• Could not conclude whether the effect related to the conversion of HOV to HOT lanes or to adding access points. |
| Jang et al. (2013) [43] | California, USA | Left, buffer-separated | Freeways: 13 routes, 246 km | 3-year data on collisions on HOV and adjacent lanes | NB regression models for PDO and injury crashes | Lane and shoulder widths, buffer width, AADT | • A wider HOV lane tends to be associated with fewer collisions.<br>• Wider shoulder width helps reduce collisions in HOV lanes.<br>• Higher AADTs in HOV and left lanes are positively related to collisions in HOV lanes but negatively—to collisions on the left lanes.<br>• Higher buffer widths have negative effects on left-lane collisions. |
| Colwill (2014) [44] | Canada, Toronto and Hamilton area | Left, buffered, limited-access | Two freeways, each with six interchanges and 2–3 sections, examined | Several years before and after HOV lanes' operation (not specified) | BA evaluation; crash trends' analysis for FI and PDO crashes | -- | • No change in FI crashes (+1%; CI −19%; +21%). A 19% increase in PDO crashes (CI +7%; +31%). A 15% increase in all crashes (sig.).<br>• A minor increase in the proportion of injury crashes, from 1% to 3%.<br>• A significant increase in rear-end collisions, particularly within the weaving and merging zones of the HOV lanes. |

**Table 1.** *Cont.*

| Source | Country, Area | Measures' Description—HOV Lanes | Road Class, Sites | Crash Data Periods | Study Design | Other Road/Traffic Features Considered | Main Findings |
|---|---|---|---|---|---|---|---|
| Srinivasan et al. (2015) [45] | USA, three States | Left, separated by painted stripes or buffer (HOV) or by buffer with flexible poles (HOT) | Urban freeway segments with HOV lanes (491 mi) or HOT lanes (27 mi) | 5-year crash data, FI and PDO, for HOV segments; 4-year crash data for HOT segments | NB regression models for FI and total crashes, with 6, 8, 10 or 12 total lanes for HOV segments and various lane numbers for HOT segments | AADT, lanes' number, left-shoulder-width, painted strip vs. buffer widths | • In all models, higher AADT and a higher number of lanes increase the crash number.<br>• In HOV lane models for 10-lane freeways, more total crashes are expected with a painted stripe compared to buffer separation, and wider buffer separation (2–3 ft) is correlated with fewer FI crashes.<br>• In HOV lane models, a wider left shoulder is associated with a decrease in the number of FI and total crashes.<br>• In HOT models, the wider the separation, the fewer crashes are expected. |
| Kim, Park (2018) [46] | California, USA | Left, buffer-separated: weaving segments | Freeways: 59 sites–weaving zones, with and without HOV lanes' access points | 3-year crash data | Descriptive analysis; explanatory models | Presence of HOV lanes' access points, length, lanes' number, AADT, lane occupancy and speed | • Weaving segments with access points showed less crashes than the counterparts without them.<br>• The higher length of the weaving segment and acceleration/deceleration sections increase crashes.<br>• Congestion-related crashes (>20% lane occupancy) prevailed on weaving segments without an access point.<br>• Low-speed differences (<10 kph) were in 78–79% of cases in pre-crash conditions in both types of sites. |
| Lee et al. (2020) [47] | USA, five States | n/a | Freeways, urban and rural; 46,955 road units | 1-year data | NB regression models for total crashes and rear-end, sideswipe, single-vehicle crash types | AADT, share of trucks, lanes' number, median and shoulder width, pavement roughness index, speed limit, area type | • HOV lane operation increased the number of total, rear-end, and side-swipe crashes when there was only one HOV lane per direction, but it had almost no effect when a segment had two or more HOV lanes.<br>• HOV lane operation decreased the number of single-vehicle crashes, regardless of the number of HOV lanes. |

**Table 1.** *Cont.*

| Source | Country, Area | Measures' Description— HOV Lanes | Road Class, Sites | Crash Data Periods | Study Design | Other Road/Traffic Features Considered | Main Findings |
|---|---|---|---|---|---|---|---|
| Yuan et al. (2021) [48] | USA, three States | Mostly, left | 11 freeways, urban and rural: 2050 mi in total | 2-year crash and traffic data | NB regression models for total crashes | Various levels of aggregation for traffic data and speed, number of lanes, speed limit | • All models indicated that HOV lane operation increases crash frequency, e.g., by 42.33% for the weekday hourly crash frequency by 21.29% in the model based on AADT in California. |
| Himes et al. (2022) [49] | USA, California and Washington | Left, HOV (or HOT) lanes, with various types of access and separation from GP lanes | Freeways, urban and rural: 543 mi in total | 5-year data | NB regression models for total crashes and multiple-vehicle crashes (in one travel direction) | AADT, speed, lanes' number, shoulder and lane widths, distances from ramps and access points, curvature, HOV lane access type and separation type, median and outside barriers' presence | • The models enable to estimate the safety of freeway facilities with continuous, buffer-separated, or barrier/pylon-separated HOV/HOT lanes while accounting for other traffic and roadway characteristics.<br>• The models indicated that barrier/pylon separation of HOV lanes decreases crash frequency compared to lane marking only; flush buffer separation increases crash frequency, but wider buffer moderates such increase.<br>• The presence of weave zones and higher speed differentials between HOV and GP lanes increase crash numbers. |

Notes: HOV—high-occupancy vehicle; HOT—high-occupancy toll; GP—general-purpose; AADT—average annual daily traffic; FI—fatal and injury; PDO—property-damage only; BA—before-after; SPF—safety performance function; EB—empirical Bayes; NB—negative-binomial; CI—confidence interval. * No control group. ** Not available.

Two recent studies from the USA applied data from several States to develop explanatory models for crashes on freeways, using various road and traffic characteristics—see Table 1 [47,48]. Both studies found that HOV lanes' operation increased the crash frequency on freeway segments. For example, models by Lee et al. [47] showed that HOV lane operation increased total, rear-end, and sideswipe crashes when there was one HOV lane per direction. Yet, almost no effect was observed when a segment had two or more HOV lanes for both directions. Yuan et al. [48] developed safety performance functions for freeways while applying data for traffic volumes with various aggregation levels; all models showed that HOV lanes' operation increased crash frequency, e.g., by 21–42%.

Being aware of the phenomenon of a different crash expectancy on freeway segments with HOV lanes related to those without HOV lanes, specific models were developed in the USA for predicting crashes on freeway segments with HOV lanes [45,49]. These models and other studies showed that design features impact crash occurrences on the HOV lanes. For example, the physical separation of HOV lanes by means of barriers or pylons decreases crash frequencies compared to a lane marking only [49]; a buffer separation may increase crash frequency compared to a lane marking [49], but a wide buffer between HOV lane and other lanes is associated with a decrease in crashes [43,45].

Furthermore, studies showed that the presence of interchange areas (weave zones) and higher speed differentials between HOV and general-purpose lanes increase crash numbers [40,41,49]. In contrast, greater HOV lane and shoulder widths are associated with lower accident frequencies [21,43,45].

At the same time, research results have not established yet whether "closed" (limited access) or "open" (continuous access) HOV lanes have a better safety performance. The practice of HOV lane settings assumes that a "closed" setting decreases lane-change maneuvers on the road segments and would reduce the risk of vehicle collisions [4,8,13]. However, accident studies, e.g., Chung et al. [39] and Jang et al. [40,41], found higher crash frequencies for "closed" HOV lanes compared to "open" settings. In line with the design expectations, Cao et al. [42] reported that adding access/egress areas to "closed" HOV lanes contributed to accident reductions. Similarly, Kim and Park [46] found that interchange areas with access/egress gates to HOV lanes were associated with lower accident rates than interchange areas without such gates.

In summary, based on previous research, it can be concluded that the installation of HOV lanes will have an adverse effect on traffic safety, i.e., an increase in accidents is expected due to an increase in lane-change manoeuvers, speed differentials between the HOV and general-purpose lanes, etc. However, most previous studies examined settings with left-side HOV lanes (near the median), which are more common in North American practice, while separate findings regarding the safety effects of right-side HOV lanes (such as in Israel) are generally lacking. Concerning the safety impacts of the design features of HOV lanes, previous research showed varying findings, which could not yet provide consistent guidance for planning HOV lanes to prevent negative safety implications. Further research on the topic is needed.

## 3. Examining Accident Changes in the Israeli Case: Data and Methods

To evaluate the safety effects associated with the introduction of HOV lanes in Israel, a before-after analysis of accident changes with a comparison group was applied. This approach is common in road safety evaluation studies [23]; it assumes that changes in accident numbers in the comparison group predict the changes that would have occurred in the treatment sites without the intervention. In this study, we examined accident changes on the treatment-group roads (with HOV lanes) in the after as opposed to the before period while accounting for accident changes on comparison-group roads in similar periods, as well as for other confounding factors such as previous accident trends, seasonality, changes in exposure and external factors (pandemic lock-downs). The latter was needed as the after period (with HOV lanes in place) included the year 2020 when pandemic lockdowns were

applied in Israel and other countries, leading to tangible fluctuations in vehicle trips and traffic exposure [50,51].

The treatment group included sections of Road #2, where the HOV lanes were introduced (see Section 1). As comparison groups, sections of other motorways from the same geographic area—the center of the country, were selected from Road #1 and Road #4. The comparison-group road sections have similar road layouts, with a physical separation between the carriageways and three travel lanes per direction; both the treatment and comparison-group roads have high daily traffic volumes, with over 96 thousand vehicles. Table 2 summarises the treatment- and comparison-group roads in the study. The *after* period was defined from the first month of full HOV lanes' operation till the end of the year 2020, i.e., between November 2019 and December 2020 (14 months); as a *before* period, a longer time period was considered, between January 2017 and September 2019 (33 months).

**Table 2.** Treatment- and comparison-group roads in the Israeli case.

| Road Group | Road # | Study Sections—Between Interchanges | Length, km | Number of Interchanges in the Study Sections |
|---|---|---|---|---|
| Treatment (T), with HOV lanes | 2 | Netania–Glilot | 21.9 | 8 |
| Comparison (C1) | 1 | Kibbutz Galuiot–Ben Shemen | 18.9 | 6 |
| Comparison (C2) | 4 | Ashdod–Dror | 53.6 | 21 |

Monthly time series of accident numbers were prepared for the treatment- and comparison-group roads, based on the Central Bureau of Statistics (CBS) accident files. Accident counts were produced for road sections and interchange areas while considering total accidents and their subdivisions by severity, day/night periods, crash types and the involvement of certain vehicle types (bus or motorcycle). Such accident subdivisions were examined to provide a comprehensive picture of changes associated with the HOV lanes' operation, and also in line with previous research [34,43,45,47,49]. Furthermore, the analysis of mobility changes indicated the impacts of HOV lanes on bus and motorcycle traffic on Road #2 [16]; thus, changes in accidents involving these vehicle types were examined respectively.

In addition, it should be mentioned that in Israel, two types of accident files are collected by the police: a "road accidents with casualties" file (termed as TD, from the words "traffic accidents" in Hebrew), with cases investigated by the police, and a "general with casualties" file (in short, "general"), with cases reported to the police but not investigated. The TD files include all severity levels of accidents and serve as a basis for the official statistics on injury accidents in the country; the "general" files include slight injury cases only, which do not satisfy the inclusion criteria of TD files. The number of records in "general" files vs. TD files is much higher, representing an 80% to 20% relation. Therefore, it is common today to include data from both files in road safety research studies [52,53] whereas TD files are considered the main data files and "general" files are complementary. Similarly, in this study, we examined data from both types of files that increased the number of data series analyzed. In total, 17 accident time series were examined in the study (see Section 4); some accident types were excluded from the formal analysis due to limited accident statistics on the treatment road.

The accident changes were estimated by fitting regression models to monthly time series of accidents, while among the explanatory variables were: time- period (before or after), site group (treatment or comparison), accident trends by period and seasonality (month of year). In addition, to control for the external factors with a substantial impact on traffic exposure, the model included the indicators of pandemic lock-down months (two periods) and of the total traffic exposure (annual vehicle-km traveled, in millions), based on values from the CBS publication [51]. We fitted negative-binomial regression models to

each series, with monthly accident counts as a dependent variable. In some cases, when the negative-binomial model could not converge, a Poisson regression model was fitted instead. GLIMMIX procedure of SAS 9.4 [54] was used to estimate the parameters of the model.

The mathematical form of the models was as follows:

$$
\begin{aligned}
\log(\lambda_{ij}) = \beta_0 &+ \beta_{type_{C1}} * I(type(j) = \text{`C1'}) + \beta_{type_{C2}} * I(type(j) = \text{`C2'}) + \beta_{type_T} \\
&* I(type(j) = \text{`T'}) + \\
\beta_{time} * time &+ \beta_{time*type_{C1}} * time * I(type(j) = \text{`C1'}) + \beta_{time*type_{C2}} * time \\
&* I(type(j) = \text{`C2'}) + \beta_{time*type_T} * time * I(type(j) = \text{`T'}) + \\
\beta_{time2} * time2 &+ \beta_{time2*type_{C1}} * time2 * I(type(j) = \text{`C1'}) + \beta_{time2*type_{C2}} * time2 \\
&* I(type(j) = \text{`C2'}) + \beta_{time2*type_T} * time2 * I(type(j) = \text{`T'}) + \\
\beta_{I\_after} * I\_after &+ \beta_{I\_after*type\_C1} * I\_after * I(type(j) = \text{`C1'}) + \beta_{I\_after*type\_C2} \\
&* I\_after * I(type(j) = \text{`C2'}) + \beta_{I\_after*type\_T} * I\_after * I(type(j) \\
&= \text{`T'}) + \\
&\beta_{I\_Lock1} * I\_Lock1 + \beta_{I\_Lock2} * I\_Lock2 + \\
\sum_{k=1}^{12} &\beta_{Mk} I(mon(i) = k) + \beta_{\log\_exposure} * \log(exposure) + offset
\end{aligned}
\tag{1}
$$

where: $\lambda_{ij}$—the number of accidents in month $i$ in series $j$; $I(type(j))$—a categorical variable indicating whether the datum belongs to the treatment- or the comparison-group road (*types* T, C1, C2, respectively); *time*—the number of observation (general over-time trend); *time2 = max(0,time-35)* indicates a change in trend in the after period; *I_after* indicates the study period ("1" for after, "0" for before period); *I_Lock1*, *I_Lock2*—indicators of the lockdown months; $I(mon(i) = k)$—a categorical variable indicating whether month $i$ is the $k$ month of the year, $k = 1, 2, \ldots, 12$; *log(exposure)*—indicator of traffic exposure, per year; *offset*—the offset variables applied to neutralize the differences in the number of days per month and among the study units; $\beta$ (all types)—the model coefficients. In the models, offset variables took the following forms: *log(days per month × length in km)*—for section accidents; *log(days per month × number of intersections)*—for junction accidents. Table 3 provides an overview of the variables included in the models.

**Table 3.** Overview of variables included in the models.

| Variable | Values | Meaning |
| --- | --- | --- |
| I(type(j)) | 0, 1 | indicator of belonging to the treatment- or comparison-group road, where *type*: T, C1 or C2 |
| I_after | 0, 1 | indicator of belonging to the study period: 1 for after, 0 for before period |
| time | 1...33, 35...48 | number of observation indicates a general over-time trend |
| time2 | 0, 1–13 | number of observation since the intervention point (HOV lanes' operation) indicates a change in trend in the after-period |
| I_Lock1 | 0, 1 | indicator of belonging to the first lockdown months |
| I_Lock2 | 0, 1 | indicator of belonging to the second lockdown months |
| I(mon(j)) | 0, 1 | indicates a month of the year (seasonal effect), where *mon*: 1–12 |
| log(exposure) | 4 numeric values | indicator of traffic exposure per year |
| offset | numeric values | Estimated as *log(days per month × length in km)*—for section accidents; *log(days per month × number of intersections)*—for junction accidents |

Two pseudo-$R^2$ measures were used to assess the quality of fit of the models. The first measure is the marginal $R^2$ of Zheng [55], which equals one minus the ratio between the sum of the squared model residuals and the variance of observed values. The second measure is the ratio between the variance of values predicted by the model and the variance of values observed, based on Nakagawa and Schielzeth [56]. To examine the significance of

both measures, we used an overall F-test, which evaluates whether all model coefficients jointly are equal to zero (the null hypothesis).

Based on the models, we estimated odds ratios (OR), which expressed the change in accident numbers in the treatment group in the after versus before period, after deducing the change in accident numbers in the comparison group and confounding factors. If the OR value is over "1", it means a worsening in safety performance, while the OR value below "1" indicates an improvement in safety in the after versus before period. If both limits of the confidence interval are greater than "1" in the first case or below "1" in the latter case, the result is significant. Figure 3 illustrates the process of data analysis in the study.

> 1: Selection of treatment and comparison-group roads

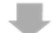

> 2: Preparation of monthly time-series of accident numbers for the treatment- and comparison-group roads (17 series)

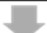

> 3: Definition of possible explanatory variables for accident models: road type (treatment or comparison); period (before and after); lockdown months; seasonality - month of the year; traffic exposure, per year; road characteristics - length, number of interchanges

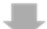

> 4: Fitting regression models to monthly time-series of accidents, with all variables defined (17 models)

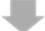

> 5: Estimating odds ratios - change in accident numbers on the treatment road in the after vs. before period, having deduced the change in accident numbers in the comparison-group and other factors

**Figure 3.** The process of data analysis in the study.

## 4. Accident Analyses of the Israeli Case: Results

To examine the impact of the HOV lanes' introduction on Road #2 in Israel, 17 accident time series were analysed in the study, as defined in Table 4. Accident changes were analysed both on road sections and in the interchange areas (termed as "junctions" in Table 4) based on accident data from two types of files (TD and "general"), as explained above. In addition, Table 4 presents accident counts observed at the study sites, in before and after periods.

The data showed (see Table 4) that, based on the main accident files (TD), severe accidents (fatal and serious) comprised between 25–40% of accidents in the study sections; in both periods, the majority of accidents were vehicles collisions (64–85%); night-time accidents represented between 20–31% of cases on road sections and 14–36% in the interchange areas, and motorcyclists were involved in a substantial share of section accidents (23–50%). Data from complementary accident files ("general") showed higher accident counts, as expected, with lower shares of night accidents (12–17% on sections, 7–24% near interchanges), and a smaller involvement of motorcyclists (9–14%). In addition, buses were involved in 3–4% of the total accidents on road sections.

**Table 4.** Accident time series analysed in the study, with accident counts observed at the study sites by period.

| No. | Time-Series Code | Accident Data File * | Accident Sites ** | Accident Type | Treatment Sites T | | Comparison Sites C1 | | Comparison Sites C2 | |
|---|---|---|---|---|---|---|---|---|---|---|
| | | | | | Before | After | Before | After | Before | After |
| 1 | 11 | TD | Sections | Total | 64 | 20 | 74 | 19 | 259 | 80 |
| 2 | 12 | TD | Sections | Fatal + serious | 16 | 8 | 25 | 6 | 76 | 32 |
| 3 | 13 | TD | Sections | Vehicle collisions | 45 | 17 | 47 | 13 | 190 | 57 |
| 4 | 14 | TD | Sections | Daytime | 50 | 16 | 55 | 14 | 196 | 55 |
| 5 | 15 | TD | Sections | Nighttime | 14 | 4 | 19 | 5 | 63 | 25 |
| 6 | 16 | TD | Sections | with MTC | 15 | 10 | 30 | 8 | 105 | 31 |
| 7 | 21 | TD | Junctions | Total | 33 | 14 | 43 | 16 | 111 | 53 |
| 8 | 22 | TD | Junctions | Daytime | 21 | 12 | 31 | 13 | 73 | 36 |
| 9 | 23 | TD | Junctions | Nighttime | 12 | 2 | 12 | 3 | 38 | 17 |
| 10 | 31 | general | Sections | Total | 675 | 151 | 320 | 88 | 1284 | 367 |
| 11 | 32 | general | Sections | Daytime | 575 | 131 | 282 | 73 | 1095 | 317 |
| 12 | 33 | general | Sections | Nighttime | 100 | 20 | 38 | 15 | 189 | 50 |
| 13 | 34 | general | Sections | with bus | 23 | 5 | 11 | 3 | 45 | 11 |
| 14 | 35 | general | Sections | with MTC | 59 | 19 | 34 | 12 | 170 | 44 |
| 15 | 41 | general | Junctions | Total | 66 | 28 | 218 | 106 | 479 | 238 |
| 16 | 42 | general | Junctions | Daytime | 50 | 26 | 194 | 93 | 422 | 202 |
| 17 | 43 | general | Junctions | Nighttime | 16 | 2 | 24 | 13 | 57 | 36 |

Notes: * *TD*—"road accidents with casualties" files; *general*—"general with casualties" files. ** *Junctions* refer to interchange areas of motorways. *MTC*—a motorcyclist.

### 4.1. A Detailed Example: Changes in the Number of Injury Accidents on Road Sections

Table A1 in Appendix A presents a negative-binomial model fitted to one time-series—monthly injury accidents on motorway sections (time-series 11 in Table 4). This model had high values of fit, of 69–70%, in terms of both pseudo $R^2$ measures [55,56] and was significant according to an overall F-test ($p < 0.05$). However, the *Type III* tests of fixed effects for the model variables showed that most variables did not have a significant impact on accident numbers, except for *time* (the general trend), which was approximately significant ($p < 0.1$) and indicated a slight decrease over time. The model coefficients indicated, in general, a higher accident expectancy on comparison-group sections related to the treatment road but a decreasing trend in accidents on the comparison roads vs. Road #2 in the after period.

Figure 4 provides visual plots, which accompanied the accident analyses in this case. Figure 4a shows monthly accident counts (raw data) on the treatment and comparison-group roads from January 2017 to December 2020. Using the model developed for the treatment and comparison-group roads, two predicted time series were produced for the after period (Figure 4b), which are:

(1) "Actual values" (*After_Ac*)—accident counts that were observed in the after period (having excluded the impact of lock-downs), which are given in solid lines, and

(2) "Expected values" (*After_Nc*)—accident counts which would occur in the after period if the trend of the before period had continued (and the treatment—HOV lanes, was not applied), which are given in dashed lines.

In both cases—the actual and expected values- Figure 4b presents the logarithm of accident numbers per day per unit (1 km of the road).

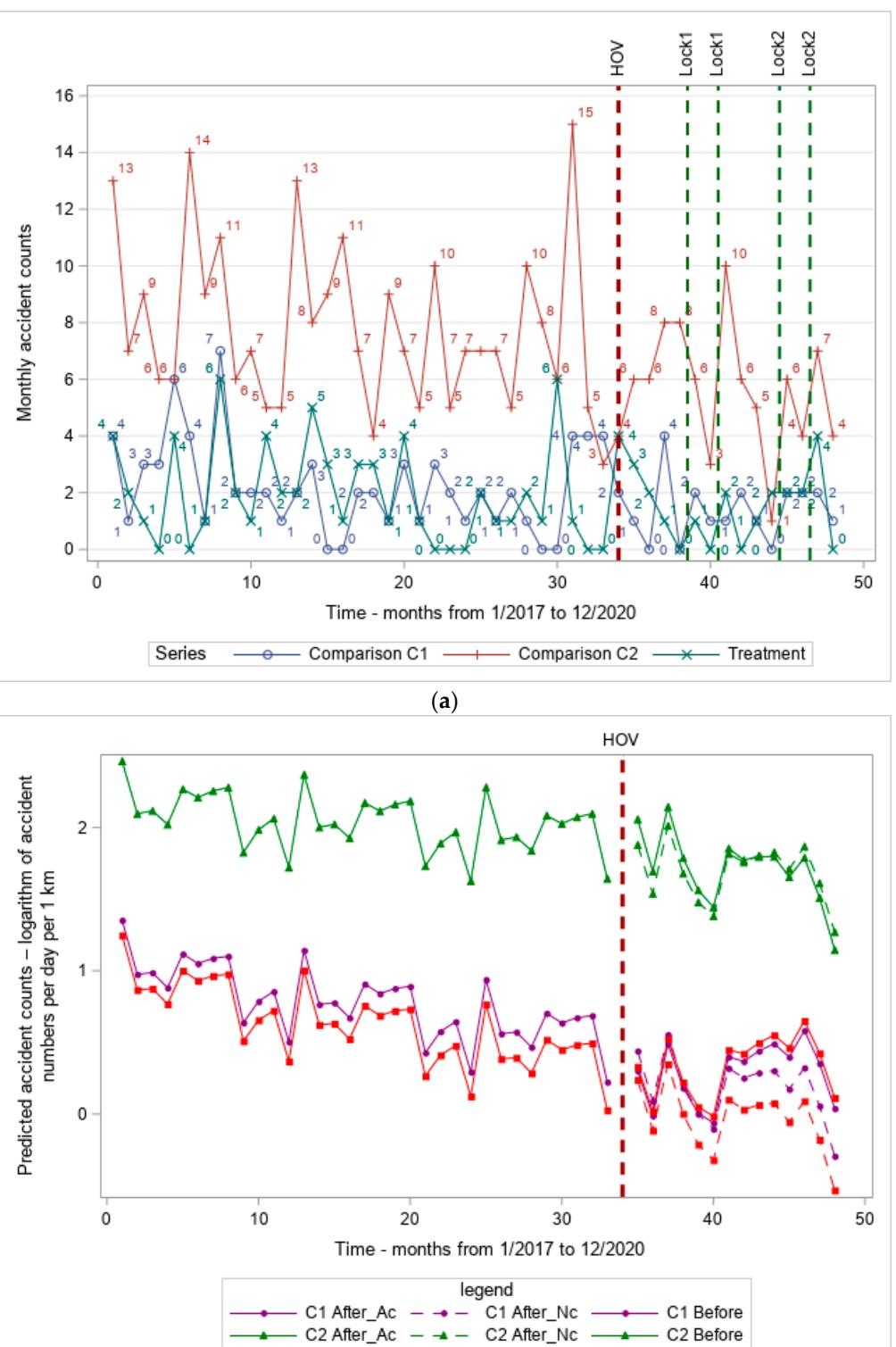

**Figure 4.** Analyses of time series of the total injury accidents on road sections: (**a**) Monthly accident counts (raw data); (**b**) Predicted accident counts for treatment (T) and comparison roads (C1, C2). Notes: *HOV*—month of introduction of HOV lanes; *Lock1*, *Lock2*—months of lock-downs. (See text for explanations on the meaning of *After_Ac* and *After_Nc*).

Furthermore, using the model, ratios were estimated as follows: *Ratio T*, which presents a ratio of the actual and expected accident numbers in the after period in the treatment group; *Ratio C1*—which presents a ratio of the actual and expected accident numbers in comparison-group C1, and *Ratio C2*—presents a similar ratio in comparison-group C2. These ratios indicate a trend in accident changes in the after as opposed to before period in each group of roads. Then, the odds ratios: *Treatment* vs. *C1* and *Treatment* vs. *C2*—show the accident changes on the road where the HOV lanes were introduced, having deduced the accident changes on the comparison-group roads at the same time-periods.

In the case of total injury accidents on motorway sections, the results showed (Table 5) that, in the after period related to before, a strong increasing trend was observed in accidents on the treatment road, whereas on comparison-group road C1 a slightly increasing trend appeared and no change was observed on comparison-group road C2. The odds ratios indicated an increase in accidents on the treatment road as opposed to both comparison-group roads, with an average addition of 31–41%, but the changes were not statistically significant, thus pointing to an increasing trend.

**Table 5.** Ratios and odds ratios estimated for total injury accidents on road sections.

| Ratio | Estimate | 95% Confidence Interval | | *p*-Value |
|---|---|---|---|---|
| Ratio T | 1.44 | 0.41 | 5.03 | 0.56 |
| Ratio C1 | 1.10 | 0.33 | 3.70 | 0.87 |
| Ratio C2 | 1.03 | 0.41 | 2.61 | 0.95 |
| Odds Ratio: Treatment vs. C1 | 1.31 | 0.35 | 4.90 | 0.68 |
| Odds Ratio: Treatment vs. C2 | 1.41 | 0.49 | 4.07 | 0.52 |

### 4.2. Changes in All Accident Time-Series Examined

Table 6 and Figure 5 provide an overview of the results of all analyses across the 17 time series examined and both comparison groups. It can be seen that, based on data from the TD files ("road accidents with casualties"), which are the main source of official accident data in Israel, mostly increasing trends were found both for accidents on road sections and in the interchange areas. More specifically, in total injury accidents on road sections, in the after period, increasing trends were observed on the treatment road related to both comparison roads, indicating an average increase by 31% and 41% (both non-significant); similarly, in severe accidents on road sections, increasing trends were indicated related to both comparison roads, with an average increase by 248% and 78% (both non-significant, with large confidence intervals). In vehicle collisions on-road sections, increasing trends were observed to the extent of 68% and 70%, on average (both non-significant). In daytime injury accidents on road sections, the change on the treatment road was minor relative to comparison road C1 (−5%) but indicated an increase relative to another comparison road C2 (+75%), both non-significant. In nighttime injury accidents on road sections, an increasing trend was observed on the treatment road relative to comparison road C1 (+176%) and a decreasing trend relative to comparison road C2 (−27%), both changes non-significant. In injury accidents with motorcyclists on road sections, increasing trends were observed related to changes in both comparison roads, to the extent of 143% and 87%, on average (both non-significant). In total injury accidents at interchange areas, increasing trends were found related to both comparison roads, with an addition of 150% and 44%, on average (both non-significant). In daytime injury accidents at interchange areas, a close to significant increase was observed relative to comparison road C1 (+488%) and an increasing trend relative to comparison road C2 (+91%). In nighttime injury accidents at interchange areas, decreasing trends were observed related to both comparison roads, indicating an average reduction of 69% and 43% (non-significant).

**Table 6.** Results of all analyses: odds ratio estimates for the treatment road vs. each comparison-group road.

| No. | Comparison-Group | Series * | *p*-Value | OR Estimate | 95% Confidence Interval | | Comparison-Group | *p*-Value | OR Estimate | 95% Confidence Interval | |
|---|---|---|---|---|---|---|---|---|---|---|---|
| 1 | C1 | 11 | 0.680 | 1.31 | 0.35 | 4.90 | C2 | 0.519 | 1.41 | 0.49 | 4.07 |
| 2 | C1 | 12 | 0.292 | 3.48 | 0.33 | 37.02 | C2 | 0.559 | 1.78 | 0.25 | 12.84 |
| 3 | C1 | 13 | 0.528 | 1.68 | 0.32 | 8.94 | C2 | 0.441 | 1.70 | 0.43 | 6.77 |
| 4 | C1 | 14 | 0.950 | 0.95 | 0.20 | 4.55 | C2 | 0.375 | 1.75 | 0.50 | 6.18 |
| 5 | C1 | 15 | 0.450 | 2.76 | 0.18 | 41.73 | C2 | 0.781 | 0.73 | 0.07 | 7.22 |
| 6 | C1 | 16 | 0.500 | 2.43 | 0.17 | 34.45 | C2 | 0.577 | 1.87 | 0.19 | 17.87 |
| 7 | C1 | 21 | 0.233 | 2.50 | 0.54 | 11.54 | C2 | 0.581 | 1.44 | 0.38 | 5.40 |
| 8 | C1 | 22 | 0.066 | 5.88 | 0.89 | 39.06 | C2 | 0.442 | 1.91 | 0.35 | 10.39 |
| 9 | C1 | 23 | 0.441 | 0.31 | 0.01 | 6.40 | C2 | 0.627 | 0.57 | 0.06 | 5.71 |
| 10 | C1 | 31 | 0.963 | 1.01 | 0.61 | 1.67 | C2 | 0.080 | 0.73 | 0.51 | 1.04 |
| 11 | C1 | 32 | 0.642 | 1.13 | 0.67 | 1.89 | C2 | 0.150 | 0.76 | 0.53 | 1.11 |
| 12 | C1 | 33 | 0.393 | 0.56 | 0.14 | 2.19 | C2 | 0.156 | 0.52 | 0.20 | 1.30 |
| 13 | C1 | 34 | 0.734 | 0.67 | 0.07 | 6.82 | C2 | 0.230 | 0.33 | 0.05 | 2.04 |
| 14 | C1 | 35 | 0.587 | 0.69 | 0.18 | 2.67 | C2 | 0.420 | 0.68 | 0.26 | 1.77 |
| 15 | C1 | 41 | 0.022 | 2.96 | 1.18 | 7.41 | C2 | 0.174 | 1.83 | 0.75 | 4.45 |
| 16 | C1 | 42 | 0.006 | 3.94 | 1.54 | 10.11 | C2 | 0.057 | 2.42 | 0.97 | 6.03 |
| 17 | C1 | 43 | 0.411 | 0.26 | 0.01 | 6.97 | C2 | 0.401 | 0.28 | 0.01 | 5.87 |

* See definitions in Table 4.

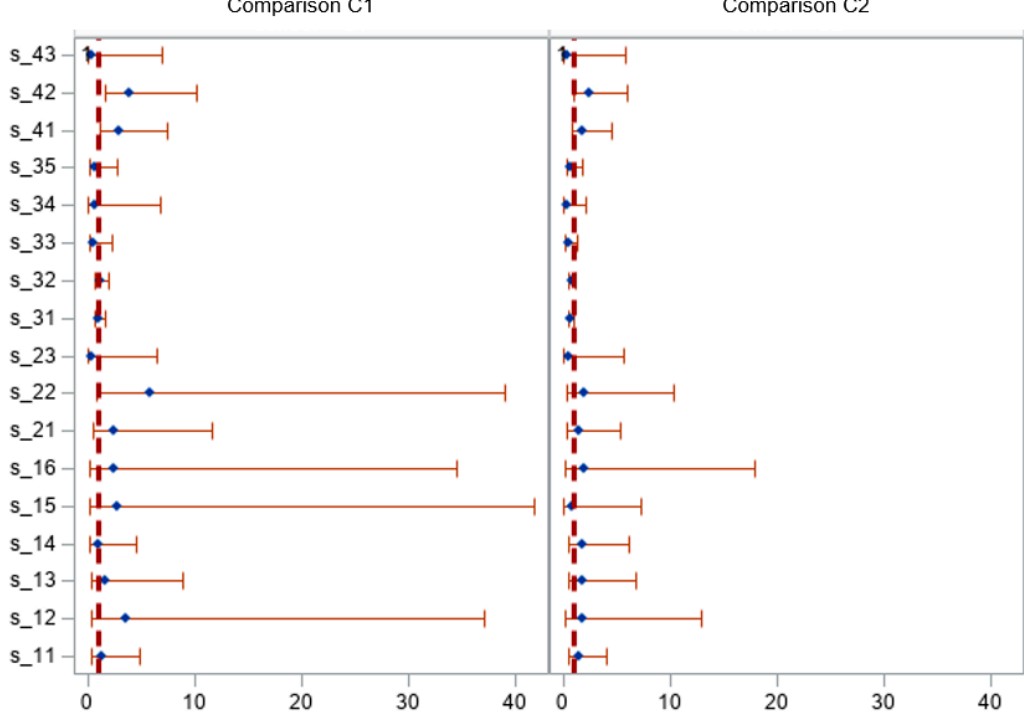

**Figure 5.** Summary of all results—accident changes in the treatment vs. comparison-group roads: odds ratio values and 95% confidence intervals, by time series. (See exact values in Table 6).

Using data from the complementary files ("general with casualties"), mixed changes were observed in section accidents but increasing trends in accidents near the interchanges. In particular, in total accidents on road sections, no change was observed relative to

comparison road C1 (+1%) and a decreasing trend relative to comparison road C2 (−27%; close to significant, $p < 0.1$). In daytime accidents on road sections, an increasing trend was found relative to comparison road C1 (+13%) and a decreasing trend relative to comparison road C2 (−24%), both non-significant. In nighttime accidents on road sections, decreasing trends were found related to both comparison roads, with an average change of 44% and 48% (both non-significant). In bus accidents on road sections, decreasing trends were observed related to both comparison roads, with a reduction of 33% and 67% on average, both non-significant. Similarly, in motorcycle accidents on road sections, decreasing trends were observed related to both comparison roads, of 31% and 32% on average, both non-significant. In total accidents at interchange areas, a significant increase was found relative to comparison road C1 (+196%) and an increasing trend relative to comparison road C2 (+83%, not significant). In daytime accidents at interchange areas, a significant increase was found relative to comparison road C1 (+294%) and a close to significant increase relative to comparison road C2 (+142%). In nighttime accidents at interchange areas, decreasing trends were observed related to both comparison roads, of 74% and 72% on average, both non-significant.

In summary, the analyses of accident changes on the motorway with HOV lanes in Israel in the first period of HOV lanes' operation showed as follows:

- In injury accidents (based on the TD files) on road sections, increasing trends were found in accident occurrences, in total accidents (with all injury levels), severe accidents, vehicle collisions and accidents involving motorcycles. However, the changes in daytime and night accidents on road sections were inconsistent in relation to different comparison-group roads.

- In injury accidents at interchange areas (based on the TD files), increasing trends were observed in total and daytime accidents, with an almost significant increase in the case of daytime accidents as compared to Road #1 (OR = 5.9, $p = 0.066$). At the same time, a decreasing trend was observed in nighttime injury accidents at interchange areas in relation to both comparison groups.

- Concerning injury accidents from the complementary files ("general with casualties"), on road sections, inconsistent changes were observed in the total and daytime accident occurrences, while in night accidents and accidents involving buses or motorcycles, decreasing trends were indicated. At the same time, regarding accidents at interchange areas, consistent increasing trends were found in the total and daytime accidents, including a significant increase as compared to Road #1 (OR = 3.0, $p < 0.05$ for total accidents; OR =3.9, $p < 0.01$ for daytime accidents) and Road #4 (OR = 2.4, $p = 0.057$ for daytime accidents). In night accidents at interchange areas, decreasing trends were observed.

- It should be noted that the decreasing trends observed in night accidents at interchange areas (according to data from both the main accident files and complementary files) are less relevant to the measure examined because HOV lanes on Road #2 operate mostly during daytime (between 6 am and 10 pm).

Furthermore, we conducted robustness checks to support our results. Following [57,58], this was done by introducing pseudo-intervention points before the actual intervention month (HOV lanes' operation in October 2019). Since only series 42 had significant OR estimates for both comparison groups (see Table 6), robustness checks were performed for this series. The expectation was that if the model were robust, we would not find any change in such pseudo-intervention points. We tested whether the model produced effects in months of pseudo-interventions, set in the 12th and 24th months, i.e., at the end of one and two years in the "before" period. No statistically significant change was found in both months, thus indicating that the effect associated with the introduction of HOV lanes was robust.

## 5. Discussion

Recent developments in the transportation system in Israel and other countries promote the use of public transport and shared trips in the form of high-occupancy vehicles. In Israel, this policy is manifested specifically in arranging HOV lanes on the main road network of the country. Previous research showed extensively that high-occupancy travel modes increase the road network capacity and allow more efficient use of existing roadways [3–7]. However, knowledge gaps are recognized regarding the safety impacts of such traffic management strategies, while safety assessments of transport policy measures are important for creating effective transportation systems [4,23,24]. Therefore, this study aspired to reduce previous research gaps regarding the safety impacts of HOV lanes and understanding the implications of modern policy developments based on a detailed review of previous international research and an evaluation of accident changes associated with introducing HOV lanes in Israel.

An international literature survey from recent decades showed that HOV lanes were frequently associated with an adverse effect on road safety. The latter exhibited, for example, in an increase in accidents following the installation of HOV lanes [21,34,35,37,44] and in accident prediction models showing that the presence of HOV lanes raises accident frequency, having controlled for other road and traffic characteristics [47,48]. The higher accident risk of roads with HOV lanes can be attributed to speed differentials between the HOV and other lanes, the reduced cross-sections of roads with HOV lanes as well as to conflicts between vehicle movements at merging and diverging areas of HOV lanes and lane changes by illegal users of the HOV lanes [21,34,35]. It should be emphasized, in this context, that international findings on the issue are currently limited to the North American experience, with mostly left-side HOV lanes used on freeways. Some studies did not find significant impacts of HOV lanes on safety, but those usually considered more specific settings, e.g., barrier-separated HOV lanes [35] or a setting including left HOV lanes combined with the use of right shoulders in peak hours [38].

Furthermore, previous research showed that HOV lanes' design features affect accident occurrences on the roads. However, various findings can be found in the literature, which are not always consistent. For example, physical separation of HOV lanes by barriers or pylons was associated with a positive safety impact compared to lane marking only, i.e., a decrease in accidents [49], while concerning the buffer separation between HOV lane and other lanes, both positive and negative safety effects were reported [43,45,49]. Similarly, previous research results have not yet established whether "closed" (limited access) or "open" (continuous access) HOV lanes have a better safety performance. Yet, some studies found [42,46] that adding access/egress areas to "closed" HOV lanes or having access/egress gates in the interchange areas was associated with lower accident rates. Overall, it appears that previous research findings are insufficient yet to provide a basis for a safety-preferable design of HOV lanes. Thus, further research on the topic is needed.

A search of previous research literature showed that separate findings regarding the safety effects of the right-side HOV lanes, like those introduced in Israel, were generally lacking. In this study, we examined the safety effects of the introduction of right-side HOV lanes on a motorway in Israel, using a before-after analysis of accident changes with a comparison group. The results showed that following the HOV lanes' operation, mostly increasing accident trends were observed both on road sections and in the interchange areas, on the treated motorway, having controlled for concurrent accident changes on the comparison roads and other confounding factors. More specifically, on road sections with HOV lanes, increasing trends were found in total injury accidents, severe accidents, vehicle collisions and accidents involving motorcycles, while at the interchange areas, increasing trends were observed in total injury and daytime accidents. For example, on road sections, in total injury accidents, an average increase of 31% and 41% was observed, related to both comparison roads and in vehicle collisions—to the extent of 68% and 70% (all changes were non-significant, indicating a trend only); in daytime injury accidents at interchange

areas, a close to significant increase was observed related to one comparison road (+488%) and an increasing trend related to another comparison road (+91%). Furthermore, at the interchange areas of the motorway with HOV lanes, consistent increasing trends were found in the total and daytime accidents based on the complementary files of slight accidents, indicating a substantial increase of 196% and 83% in total accidents and 294% and 142% in daytime accidents, related to both comparison groups (some changes were significant). In summary, the introduction of HOV lanes on the Israeli motorway was associated with mostly increasing trends in accident occurrences, particularly in the daytime, i.e., in the hours of HOV lanes' operation.

In general, the accident changes observed under the Israeli conditions aligned with the international experience [21,23,37,44], i.e., the expectation of an adverse safety effect of HOV lanes. However, most evaluation results in the current study were non-significant, indicating trends only in accident changes. The limitations of the current research lie in the relatively short after period of accident occurrences included in the analyses (14 months) and the focus on one motorway where the HOV lanes were introduced. Further research on the topic would be useful to consider longer periods of HOV lanes' operation and more evaluation sites—motorway sections where HOV lanes were applied. In addition, further research should draw more attention to the safety impacts of the design features of HOV lanes, such as the use of right-side or left-side HOV lanes, the use of closed or open HOV lanes, the type of HOV separation from general-purpose lanes—for example, separation by pylons' installation, setting a flat buffer or lane marking only; HOV lane arrangements near interchanges—for example, the length and frequency of access/egress gates along HOV lanes and their combinations with merging and diverging areas of the interchanges. Knowledge of the safety impacts of such features is essential to provide a better background for the future design of HOV lanes.

Furthermore, future research may include behavior observations of lane change maneuvers and vehicle speeds, particularly near interchanges but also on road sections with HOV lanes, to understand better the reasons for higher accident frequencies associated with HOV lanes. A few previous studies examined vehicle behaviors in this context [31–33], to compare the design alternatives examined. A promising research direction can be in exploring speed differentials between the HOV and other lanes [21,35,49] under various road design and traffic conditions or in developing indicators of accident risks based on behavior analyses, as it was demonstrated, for example, in [59].

## 6. Conclusions

Recognizing the positive impacts of HOV lanes on the effective use of road network capacity, this study examined the safety impacts of HOV lanes based on recent international research findings and Israeli data analyses. The international findings showed that the installation of HOV lanes has an adverse effect on road safety, i.e., an increase in accident risk is expected. However, the findings were limited to North American experience, with mostly left-side HOV lanes in use. The introduction of right-side HOV lanes on the Israeli motorway was associated with mostly increasing trends in accident occurrences, particularly at interchange areas and in the daytime, i.e., in the hours of HOV lanes' operation. Thus, the accident changes observed under the Israeli conditions, although for right-side HOV lanes, also indicated negative safety impacts of HOV lanes.

Hence, in general, adverse safety effects should be expected and accounted for in future planning of the extension of HOV lanes in the country. While planning the application of HOV lanes, possible adverse safety effects should be considered in the range of all expected impacts of this solution, such as reduced congestion, travel time-savings, better reliability of high-occupancy travel modes, etc. In a more general sense, understanding the safety implications of the transport policy should become an inherent part of any development in the transportation system [24].

In the future, more extended evaluation studies on the safety impacts of HOV lanes are needed to provide a solid basis for safer design solutions for HOV lanes under various

road and traffic conditions. Both accident and vehicle behavior research can be useful in this context.

**Author Contributions:** Conceptualization, V.G.; literature survey, V.G.; methodology, V.G. and E.D.; software, E.D.; validation, E.D. and V.G.; formal analysis, E.D.; resources, V.G.; writing—original draft preparation, V.G.; writing—review and editing, V.G. and E.D.; visualization, V.G.; project administration, V.G.; funding acquisition, V.G. All authors have read and agreed to the published version of the manuscript.

**Funding:** The research presented in this paper was partially funded by the Israeli Smart Transportation Research Center (ISTRC).

**Institutional Review Board Statement:** Not applicable.

**Informed Consent Statement:** Not applicable.

**Data Availability Statement:** The database prepared in this study can be provided upon request.

**Acknowledgments:** The authors want to thank Anna Korchatov for her assistance in accident data preparation for this study. Appreciation is extended to Shalom Hakkert, from the Transportation Research Institute of the Technion, for his participation in the study conceptualization and valuable comments on its findings, and to Tomer Toledo from the Technion, for the assistance in funding acquisition and with administrative issues of this study.

**Conflicts of Interest:** The authors declare no conflict of interest.

## Appendix A

**Table A1.** Negative-binomial regression model for injury accidents on road sections.

| Variable | Estimate | Standard Error | DF | t Value | Pr > |t| |
|---|---|---|---|---|---|
| Intercept | −17.5814 | 30.5611 | 89.02 | −0.58 | 0.5665 |
| time | −0.02268 | 0.01569 | 42.37 | −1.45 | 0.1556 |
| type: C1 | 0.2500 | 0.3500 | 36.38 | 0.71 | 0.4795 |
| type: C2 | 0.3084 | 0.2876 | 36.35 | 1.07 | 0.2906 |
| type: T | 0 | . | . | . | . |
| time * type C1 | 0.002820 | 0.01954 | 36.11 | 0.14 | 0.8861 |
| time * type C2 | 0.01246 | 0.01597 | 36.1 | 0.78 | 0.4405 |
| time * type T | 0 | . | . | . | . |
| time2 | 0.04257 | 0.07111 | 48.9 | 0.60 | 0.5522 |
| time2 * type C1 | −0.00636 | 0.09071 | 40.35 | −0.07 | 0.9445 |
| time2 * type C2 | −0.06589 | 0.07099 | 40.27 | −0.93 | 0.3589 |
| time2 * type T | 0 | . | . | . | . |
| I_after | 0.09142 | 0.6290 | 47.44 | 0.15 | 0.8851 |
| I_after * type C1 | −0.2296 | 0.8200 | 42.9 | −0.28 | 0.7809 |
| I_after * type C2 | 0.08700 | 0.6415 | 43.01 | 0.14 | 0.8928 |
| I_after * type T | 0 | . | . | . | . |
| I_Lock1 | −0.1881 | 0.3803 | 91.05 | −0.49 | 0.6221 |
| I_Lock2 | 0.3344 | 0.3751 | 98.85 | 0.89 | 0.3749 |
| mon: 1 | 0.6300 | 0.2573 | 104.7 | 2.45 | 0.0160 |
| mon: 2 | 0.3741 | 0.2737 | 111.2 | 1.37 | 0.1744 |

**Table A1.** *Cont.*

| Variable | Estimate | Standard Error | DF | t Value | Pr > |t| |
|---|---|---|---|---|---|
| mon: 3 | 0.3036 | 0.2798 | 110.7 | 1.09 | 0.2802 |
| mon: 4 | 0.2509 | 0.2849 | 110.7 | 0.88 | 0.3803 |
| mon: 5 | 0.4742 | 0.2643 | 111.1 | 1.79 | 0.0755 |
| mon: 6 | 0.4596 | 0.2672 | 111 | 1.72 | 0.0882 |
| mon: 7 | 0.4828 | 0.2655 | 110.9 | 1.82 | 0.0717 |
| mon: 8 | 0.5172 | 0.2650 | 110.6 | 1.95 | 0.0535 |
| mon: 9 | 0.1066 | 0.3034 | 106.6 | 0.35 | 0.7260 |
| mon: 10 | 0.2423 | 0.3154 | 113.5 | 0.77 | 0.4439 |
| mon: 11 | 0.3637 | 0.2664 | 85.8 | 1.37 | 0.1758 |
| mon: 12 | 0 | . | . | . | . |
| log(exposure) | 1.0642 | 2.7779 | 89.15 | 0.38 | 0.7026 |

Model fit statistics: $-2$ Res Log Pseudo-Likelihood = 312.36; Generalized Chi-Square = 115.00; Generalized Chi-Square/DF = 1.00. Pseudo $R^2$ measures: 69.3% based on [53], 70.4% based on [54]; significance test: F-value = 1.75, $p = 0.0385$.

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
