# Peer review of "Examining the Safety Impacts of High-Occupancy Vehicle Lanes: International Experience and an Evaluation of First Operation in Israel"

_sustainability, doi:10.3390/su151813976_

Round 1

Reviewer 1 Report

The article presents investigation on safety analysis at HOV lanes. The first stage of research was applied to review available literature published from 2001 to 2022. Most publications address North America HOV lanes and left-side HOV lanes. In the next stage of research, authors gathered and analyzed data from Israel to investigate right-side HOV lanes, which are used in Israel. Equation number (1) requires reformulation and is unreadable in present form.

Author Response

please see a file attached

Reviewer 2 Report

1.      The analysis in the abstract is mostly qualitative, and the quantitative analysis results are lacking.

2.      Some studies in the literature have developed approaches that estimate the accident risk coefficient depending on vehicle speed. The literature review should also include such studies. For example, see https://doi.org/10.1049/iet-its.2019.0322.

3.      How do the authors prove that the picture given in Figure 1 is the appearance of HOV lanes in Israel?

4.      In reference citations, e.g., remove statements. For example: [e.g., 23].

5.      A "Flow diagram to identify the sources for the review." commendable to be given. However, it is difficult to understand and quite complex. It can bring new breath to the literature review, but it should be made understandable. Maybe converting it to a Table will make it meaningful.

6.      Please pay attention to the reference numbering in the text. It has a jumbled sequence and is very difficult to follow. It should be in sequential order.

7.      There is no need to use the term "international literature." Literature is already universal.

8.      What is the x-scale (time) unit in Figure 3? Seconds, minutes, hours, etc. Similarly, what is the unit of the y-scale in Figure 3? In addition, the labels (numbers) presented on the curves in Figure 3(a) are difficult and confusing to follow. There must be another way to present labels.

9.      Make all of the horizontal error bars visible in Figure 4. It is also recommended to include the numerical values of the error bars.

10.  In order to ensure the validity of the model used, a robustness test should be performed, and the results should be supported.

11.  Deep and mechanistic discussions are required to explain the results obtained. In the section "Results and discussion", the study's comparative analysis with others must be thoroughly discussed. The findings of the paper should be described with quantitative analysis results, such as percent or numerical results.

12.  It seems meaningful to give a brief summary of the general results from previous studies in the conclusion section. However, the results section should be expanded to present the numerical values of the model results used in the paper. In addition, the conclusion section is missing some perspectives related to future research work. Also, in the conclusion, readers now expect conclusive results and evidence. Therefore, there is no need to make new definitions in this section. For example: reference [55].  

Moderate editing of English language required

Author Response

please see a file attached

Reviewer 3 Report

-          The literature review only includes studies from North-American countries. Are there any studies from other countries that could provide valuable insights into the safety impacts of HOV lanes?

-          The paper mentions that safety concerns were raised regarding the operation of HOV lanes in Israel, particularly in merging and diverging areas near interchanges. Could the study provide more detailed information on these specific safety concerns?

-          The study conducted before-after comparisons of accident changes with comparison groups. How were the comparison groups selected and what factors were considered in the selection process?

-          The study conducted regression models to analyze the monthly time-series data of accident types. Could the authors provide more information on the variables included in the regression models and how they were selected?

-          The paper mentions that the introduction of HOV lanes in Israel led to increasing accident trends, particularly at interchange areas and in daytime. Could the authors provide a more detailed analysis of these trends and their potential causes?

-          The study suggests that adverse safety effects should be expected and accounted for in future planning of HOV lanes. Could the authors provide some recommendations on how these safety effects can be mitigated or reduced?

-          The paper mentions the need for further research to explore design features of HOV lanes that can reduce their negative safety implications. Could the authors provide some suggestions for specific design features that should be explored in future research?

-          How have recent developments in transportation systems in Israel focused on promoting the use of public transport and shared trips through high-occupancy vehicles (HOV)?

-          What are some of the factors that contribute to the higher accident risk associated with roads with HOV lanes?

-          What are the findings of this study regarding the safety effects of right-side HOV lanes in Israel?

-          A flow chart of the developed method is needed to make it more understandable.

-          Please ensure consistency in using different key terms.

-          Identifying key factors should not be a part of the literature review. It should be a part of the result analysis. Literature review analyses the closely related papers to identify research gaps.

-          The result analysis should be improved based on the unique findings, interesting insights, and how these results will be useful to the practice.

-          The conclusion section can be revised considering unique findings, contributions, limitations, and future research directions.

-          Check the citations and references (one by one) if there is any missing information. Citations and references must be 100% accurate.

The English of the paper is not clear in several parts, and some parts are not clear enough to understand the authors' idea. The English should be improved and the grammatical mistakes should be corrected.

Author Response

please see a file attached

Reviewer 4 Report

This paper examines the safety of right-side High Occupancy Vehile (HOV) lanes, which are being newly introduced to Israel, since mostly the data from USA and Canada, where left-side HOVs are in use, are available. The study concludes that the HOVs in Israel increase number of accidents.

The structure of the paper is good. The related work is quite extensive and is placed to a separate section, which increaces legibility of both Sections 1 and 2.

The entire approach is well designed and well described. The approach is based on before-after analysis of accidents changes with a comparison group. The authors use this approach partially due the cooincidence of their research with COVID situation, which temporarily, yet significantly affected the (road) traffic around the world. Neglecting this would negatively influence the results of the research. The authors used 17 types of accidents, while some of them weere not included due to limited amount of data.

The mathematical form of the model is quite confusing, as the equation (see Eq. (1)) is very long. It is difficult to say how to improve its legibility, maybe to use some higher level form (with variables representing several members of the equation). Each variable can be then expressed by its own (shorter) equation.

The results are described in a great detail. The discussion and the conclusions seem to be sound.

Future work is missing in the Conclusion section and should be added. Two sentences would suffice.

The figures are appropriate and of sufficient quality.

The references seem to be relevant and up-to-date.

The English is good, but there are some typos and errors. Hence, proofreading by a grammar-skilled native speaker is encouraged.

Author Response

please see a file attached

Round 2

Reviewer 2 Report

The authors have improved their paper significantly and have responded to many of my previous concerns. They developed the technical and scientific value of paper. It has now become available to be published in its current form.

I recommend the publication of the revised manuscript, sustainability-2578890, in the Sustainability. 

Reviewer 3 Report

the authors provided a point to point revision and in conclusion, the paper has key elements necessary for a successful publication.